

# Impact of selected risk factors on motor performance in the third month of life and motor development in the ninth month

Ewa Gajewska[1], Jerzy Moczko[2], Mariusz Naczk[3], Alicja Naczk[4] and Magdalena Sobieska[5]

[1] Chair and Clinic of the Developmental Neurology, Poznan University of Medical Sciences, Poznan, Poland
[2] Department of Computer Science and Statistics, Poznan University of Medical Sciences, Poznan, Poland
[3] Institute of Health Sciences, Collegium Medicum, University of Zielona Gora, Zielona Gora, Poland
[4] Department of Physical Education and Sport, Faculty of Physical Culture in Gorzow Wielkopolski, University School of Physical Education in Poznan, Gorzow Wielkopolski, Poland
[5] Department of Rehabilitation and Physiotherapy, Poznan University of Medical Sciences, Poznan, Poland

Corresponding author
Ewa Gajewska,
ewagajewska1011@gmail.com

## ABSTRACT

**Background:** Proper motor development can be influenced by a range of risk factors. The resulting motor performance can be assessed through quantitative and qualitative analysis of posture and movement patterns.

**Methods:** This study was designed as the cohort follow-up of the motor assessment and aimed to demonstrate, in a mathematical way, the impact of particular risk factors on elements of motor performance in the 3rd month and the final motor performance in the 9th month of life. Four hundred nineteen children were assessed (236 male and 183 female), including 129 born preterm. Each child aged 3 month underwent a physiotherapeutic assessment of the quantitative and qualitative development, in the prone and supine positions. The neurologist examined each child aged 9 month, referring to the Denver Development Screening Test II and assessing reflexes, muscle tone and symmetry. The following risk factors were analyzed after the neurological consultation: condition at birth (5th min Apgar score), week of gestation at birth, intraventricular hemorrhage, respiratory distress syndrome, and the incidence of intrauterine hypotrophy and hyperbilirubinemia determined based on medical records.

**Results:** A combination of several risk factors affected motor development stronger than any one of them solely, with Apgar score, hyperbilirubinemia, and intraventricular hemorrhage exhibiting the most significant impact.

**Conclusions:** Premature birth on its own did not cause a substantial delay in motor development. Nonetheless, its co-occurrence with other risk factors, namely intraventricular hemorrhage, respiratory distress syndrome, and hyperbilirubinemia, notably worsened motor development prognosis. Moreover, improper position of the vertebral column, scapulae, shoulders, and pelvis in the third month of life may predict disturbances in further motor development.

# INTRODUCTION

Certain conditions must be met for the motor development to proceed correctly, such as correct genetic imprint and adequately functioning central nervous system, comprising normal mental development, fully-working senses, and proper motor outcomes (*Vojta & Peters, 2007*; *Illingworth, 2012*; *Hadders-Algra, 2004*). Even if in later months of life, the motor performance differs due to environmental (familial, cultural, and socio-economic factors), early motor performance seems to follow a very similar pattern in all regularly developing children (*Hadders-Algra, 2018*).

Adequate cognitive development is essential in allowing the child to perform a given task or function due to motivation: the natural desire to explore the world (*Adolph & Hoch, 2019*; *Newell, 1986*). Several authors previously discussed that various risk factors influence proper motor development. The most crucial biological risk factors include the type of delivery (*Lee, Han & Lee, 2012*), low 5th-min Apgar scores (*Bulbul et al., 2020*), respiratory distress syndrome (RDS) (*Janssen et al., 2008*), intrauterine hypotrophy, hyperbilirubinemia (*Wusthoff & Loe, 2015*), and intraventricular haemorrhage (IVH) (*Bulbul et al., 2020*; *Tatishvili et al., 2010*; *Wildin et al., 1995*).

Motor development follows a specific pattern, conserved across the human population, permitting the development of homogeneous assessment methods. Such evaluation should be based on quantitative and qualitative components appropriate for a given developmental stage (*Gajewska et al., 2013*; *Gajewska, Sobieska & Moczko, 2014*). Furthermore, it should allow detecting delays or abnormalities in motor skills provides a basis for therapy plan development and makes it possible to predict further motor development. There is no consensus on a "gold standard" on uniform functional assessment in pediatric physiotherapy in Poland. Similarly, there currently needs to be a unified methodology for neurological evaluation. General movements assessment (GMs) offers a notable degree of cerebral palsy predictability in Poland and worldwide. Regrettably, defining therapeutic goals based on this method can be challenging. The Alberta Infant Motor Scale (AIMS) is an excellent assessment tool but has yet to be validated in Poland. However, it does not sufficiently capture the qualitative issues, which often tend to be crucial during therapy (*Boonzaaijer et al., 2021*). Furthermore, the Test of Infant Motor Performance (TIMP) was designed to detect developmental delays in children younger than 4 months (*Peyton, Schreiber & Msall, 2018*). It is reported in the literature that a treatment plan can be prepared based on this scale, but it does not allow to predict which child will develop cerebral palsy accurately (*Tripathi et al., 2022*). However, after a thorough analysis of the test and an attempt to use it, based on the experience of many physiotherapists, we believe that it does not meet the standard requirements, as it is challenging to identify the main problem and assign an appropriate therapy on its basis.

Previous publications have shown that the most appropriate moment for the assessment to predict further motor development in the third month of life (*Vojta & Peters, 2007*; *Hadders-Algra, 2004*; *Gajewska et al., 2013*; *Gajewska, Sobieska & Moczko, 2014*). In line

with this idea, a quantitative and qualitative evaluation scale for motor development in the third month of life was presented and validated through inter and intraobserver reliability analysis. Furthermore, it was compared with a neurological evaluation method based on the Denver Developmental Screening Test II, commonly used in Poland, supplemented with reflexes, muscle tone, and symmetry/asymmetry testing. The results regarding its usefulness were published in several scientific publications (*Gajewska et al., 2013*; *Gajewska, Sobieska & Moczko, 2014*; *Gajewska et al., 2015a*). The assessment of spontaneous motor activity involves a thorough analysis of posture and movement patterns. The quantitative evaluation determines the global pattern: a child's most advanced function (a milestone). While the movement does not need to be performed flawlessly, it is sufficient for the child to manifest it in any way or at least strive to achieve it (*Vojta & Peters, 2007*). The qualitative assessment focuses on the elements that make up one global movement pattern, accurately analyzing its kinesiological content (*Vojta & Peters, 2007*). This assessment method allows for planning the therapy aimed at individual deficits and makes it possible to predict further motor development (*Gajewska, Sobieska & Moczko, 2014*; *Gajewska et al., 2015a*). Earlier publications mainly intended to demonstrate a proper analysis of the qualitative assessment of motor development and determine predictors of further motor development, for example, satisfactory quality in the 3rd month, which is necessary to achieve high support in the 6th month (*Gajewska, Sobieska & Moczko, 2014*) independent sitting and crawling in the 9th month (*Gajewska et al., 2015b*); the position of the pelvis in the 3rd month which is crucial for the reaching of the crawl position in the 7th month (*Gajewska et al., 2021b*). However, we know that many risk factors can affect motor development and decided to evaluate them using a qualitative assessment-based scale. We decided to limit our analysis to the known biological risk factors which are supposed to have the most substantial impact on motor development. Many other factors may influence motor performance and need further analysis. Risk factors that affect motor development: maternal gestational and obstetric history (planned pregnancy, type of delivery; the number of prenatal consultations; use of medication and gestational intercurrence); features and biological risk of the baby: (sex, mechanical ventilation, feeding difficulty, socio-demographic factors), carrier and level of maternal schooling, number of children and people in the house, psycho-social issues (the family regular participation and presence of psychic risk) (*Crow, Deakin & Longden, 1975*). Out of those factors, the biological ones are regarded as the most crucial for motor development.

Hence, this study aims to statistically determine the impact of particular risk factors on elements of motor performance in the 3-month-old infants, as well as final motor performance in the 9th month of life.

## MATERIALS AND METHODS

### Participants

The study group consisted of children with no symptoms of impaired motor development, born at term or preterm (between weeks 28 and 37 of gestation), and children referred to the Clinic of Neurology for a periodic development assessment by a general practitioner, a

pediatrician, or due to parents' concerns (weak head control during traction response or suspicion of delayed development). Data were collected as previously described (*Gajewska et al., 2021a*).

The entire study population included 419 children: 236 boys and 183 girls, with 290 of them born at term and 129 preterms. The average birth week of the included infants was 38 ± 3 (born at term 40 ± 1 weeks; preterm 34 ± 3 weeks), and the mean body weight was 3,100 ± 814 g (born at term 3,462 ± 505 g; preterm 2,282 ± 788 g). Preterm children were assessed at the corrected age (*Pin et al., 2009*).

Exclusion criteria comprised genetic or metabolic disorders, severe congenital disabilities, or extreme preterm birth (below the 28th week of gestation). Moreover, no children with microcephaly or macrocephaly were included in the study.

## Procedure

The study was designed as a cohort follow-up of the motor assessment. The examination was performed at the clinic of the Greater Poland Center for Child and Adolescent Neurology and the child clinic in the years 2018–2021. The calculation of the sample size, regarding the number of newborns per year in the area, showed the required sample size of 383. We decided to gather even more population. The calculation was done using the Statistica.pl software; assuming the population of the region, where the study was performed, and the expected percentage of children affected with motor disturbances. Children whose parents/guardians asked for a screening test to exclude or confirm developmental abnormalities were recruited in the Western part of Poland. Children referred for functional assessment by pediatricians or neurologists were also included. All consecutive children aged 12–16 weeks, who presented at the Clinic were included if they did not meet the exclusion criteria. Exclusion criteria were genetic or metabolic disorders, severe birth defects, or extreme preterm birth (below 28 gestation week), microcephaly or macrocephaly. Demographics related to a child's health are taken from a medical record book or discharge summaries.

Altogether 450 were assessed and 21 were excluded due to: genetic disease (13); spina bifida (2), brachial plexus (2); metabolic disease (1); hydrocephalus (3).

The prospective statistical analysis aimed to depict the risk factors that at most influenced the motor performance in 3-month-old infants, expressed as sum of properly performed elements in the prone and supine positions, and affected at most the achieving of the proper motor development at the age of 9 months. As the second step in the statistical analysis, the association of the particular risk factors with the particular elements of the motor performance assessed in the prone and supine positions was measured, as well as the association of the particular elements with the achieving of the proper motor performance in the 9th months of life was measured.

In all children, a physiotherapeutic qualitative assessment of motor performance at 3 months (completed 3 months but before completing 4 months) was performed in the prone and supine positions, as presented in previous publications (*Gajewska et al., 2013*; *Gajewska, Sobieska & Moczko, 2014*; *Gajewska et al., 2021a*). During the assessment, a child was placed on the rehabilitation table, without clothes (so that the qualitative features

could be accurately assessed), in a warm room (temperature 21–22 °C), full-fed and well-rested (child in a state of vigilance, active, non-sleepy, non-hungry (1 h after feeding)). The examination lasted 10 to 15 min (*Gajewska et al., 2013*; *Gajewska, Sobieska & Moczko, 2014*).

In the prone position, the assessment involved: isolated head rotation; arm in front; forearm in an intermediate position; elbow outside of the line of the shoulder; palm loosely open; thumb outside, spine in segmental extension; scapula situated in medial position; pelvis in an intermediate position; lower limbs situated loosely on the rehabilitation table; foot in an intermediate position. In the supine position, the assessment involved: head symmetry; spine in extension; shoulder in a balance between external and internal rotation; wrist in an intermediate position; thumb outside; palm in an intermediate position; pelvis extended; lower limb situated in moderate external rotation and lower limb bent at the right angle at hip and knee joints; foot in an intermediate position—lifted above the rehabilitation table. Both sides were assessed for symmetrical parts of the body to exclude asymmetry. Each element was evaluated as 0-performed only partially or entirely incorrectly, 1-performed correctly. Each assessed element had to be observed at least three to four times during the test. The result was expressed as a sum of points (0–15 for prone and 0–15 for the supine position), and is presented in Table 1.

The qualitative assessment included 15 elements in prone and supine positions (Tables 2 and 3, respectively).

A neurologist examined the infants at 9 months of age. The evaluation was based on the Denver Development Screening Test II (DDST II), the assessment of reflexes, muscle tone (hypotonia or hypertonia), and symmetry (*Touwen, 1976*; *Ślenzak & Michałowicz, 1973*). The proper performance was the basis to qualify a child as adequately developed for the 9th month of life. In case of irregularities, the neurologist indicated the maximum level of motor development achieved by a child. In the case of children in which cerebral palsy (CP) was suspected, the final diagnosis could be confirmed at 18 months of age (Fig. 1). Two neurologists participated in the study, their conformity reached very high level (Cohen's *kappa* = 1) (*Gajewska, Sobieska & Moczko, 2014*).

Previously, this type of examination was used in the assessment of children aged 3 months and the comparison between physiotherapeutic and neurological assessment showed high agreement, with high conformity coefficients ($z = -5.72483$, $p < 0.001$) (*Gajewska, Sobieska & Moczko, 2014*).

The following risk factors were analyzed after the neurological consultation: condition at birth (5-min Apgar score) Apgar in the 5th min is commonly regarded as the most predictive value (*Lai, Flatley & Kumar, 2017*), week of gestation at birth, intraventricular hemorrhage (IVH) in some children, brain sonography was performed after birth, while all infants were subjected to this examination at the second month of corrected age; intraventricular hemorrhages (IVH) were classified into four grades of severity, as indicated by *Papile et al. (1978)*, respiratory distress syndrome (RDS), and the incidence of intrauterine hypotrophy and hyperbilirubinemia determined based on medical records.

The study was approved by the Research Ethics Committee of Poznan University of Medical Sciences and registered under no. 22/10 (07-01-2010). Children recruited for the

**Table 1 Risk factors, motor performance at 3<sup>rd</sup> month and final motor performance at 9<sup>th</sup> month.** Qualitative assessment was expressed as the sum of particular elements in the prone and supine positions (median and quartiles; max = 15). The final assessment was performed by the neurologist at the age of 9 months and is expressed as a number of children who reached the given level of motor performance. If the number of children in a group was lower than seven, particular values instead of median and quartiles were given. The statistical difference is for each comparison expressed as follows: Mann-Whitney U test (z=; p=); Hodges-Lehman test (two-sided exact $p$); theta value with confidence intervals. Statistically significant differences are marked in bold. CP, cerebral palsy; IVH, intraventricular hemorrhages; RDS, respiratory distress syndrome.

| The risk factors | Quality in the prone position Median (Q25–Q75) | Quality in the supine position Median (Q25–Q75) | Final assessment at the age of 9 months | | | | |
|---|---|---|---|---|---|---|---|
| | | | Suspected CP | 6<sup>th</sup> month | 7<sup>th</sup> month | 8<sup>th</sup> month | 9<sup>th</sup> month |
| Sex: girls; $n$ = 183 | 15 (8–15) | 15 (9–15) | 7 | 7 | 17 | 10 | 142 |
| boys; $n$ = 236 | 15 (9–15) | 15 (11–15) | 3 | 12 | 26 | 8 | 187 |
| Significance of the difference, boys *vs* girls | 0.65; 0.509 0.5095 0.0000 (0–0) | 0.98; 0.324 0.3247 0.0000 (0–0) | | | | | |
| No risk factors, $n$ = 254 gestation age 39.5 ± 1.1 | 15 (10–15) | 15 (13–15) | 1 | 9 | 24 | 8 | 212 |
| One risk factor, $n$ = 33 gestation age 36.7 ± 2.6 | 15 (9–15) | 15 (8–15) | 0 | 1 | 3 | 3 | 26 |
| Significance of the difference, born at term, no risk factors *vs* one risk factor | 0.99; 0.318 0.3199 0.0000 (0–0) | 0.79; 0.42 0.4262 0.0000 (0–0) | | | | | |
| Preterm, no other risk factors, $n$ = 80 gestation age 35.4 ± 1.8 | 15 (10–15) | 15 (11–15) | 3 | 1 | 5 | 3 | 68 |
| Significance of the difference, no risk factors, born at term *vs* preterm | 3.15; 0.002 0.9750 0.0000 (0–0) | 3.54; 0.004 0.5081 0.0000 (0–0) | | | | | |
| Preterm+1 risk factor, $n$ = 25 gestation age 33.5 ± 2.4 | 8 (2–15) | 11 (6–15) | 1 | 4 | 3 | 3 | 14 |
| Preterm+2 risk factors, $n$ = 18 gestation age 31.7 ± 3.1 | 5 (0–12) | 6 (0–15) | 3 | 2 | 6 | 1 | 6 |
| Preterm+3 risk factors, $n$ = 6 gestation age 31.7 ± 3.1 | 0 (0–5) | 0 (0–4) | 1 | 1 | 2 | – | 2 |
| Significance of the difference, preterm, one risk factor *vs* two risk factors | 1.42; 0.156 0.1554 2.0000 (0–8) | 1.26; 0.208 0.2079 2.0000 (0–8) | | | | | |
| Significance of the difference, preterm, one risk factor *vs* three risk factors | 1.92; 0.055 0.3164 0.0000 (0–9) | **2.69; 0.007** 0.0724 0.0000 (0–13) | | | | | |
| Significance of the difference, preterm, two risk factors *vs* three risk factors | 1.02; 0.307 0.2786 0.0000 (0–0) | 1.80; 0.072 0.0754 0.0000 (0–0) | | | | | |
| Particular risk factors, as single or in combinations | | | | | | | |

| The risk factors | Quality in the prone position Median (Q25–Q75) | Quality in the supine position Median (Q25–Q75) | Final assessment at the age of 9 months | | | | |
|---|---|---|---|---|---|---|---|
| | | | Suspected CP | 6th month | 7th month | 8th month | 9th month |
| Apgar 5th min | 7;4 | 4;4 | 1 | – | – | – | 1 |
| 0–3, n = 2 | 8 (0–11) | 12 (0–15) | 2 | 1 | – | 2 | 9 |
| 4–7, n = 22 | 15 (9–15) | 15 (11–15) | 7 | 17 | 43 | 16 | 320 |
| 8–10, n = 394 | | | | | | | |
| Significance of the difference: Apgar 5th minute 4–7 vs 8–10 | **3.28; 0.001 0.0009** −4.0000 (−7 to 0) | **3.00; 0.003 0.003** −2.000 (−3 to 0) | | | | | |
| IVH, n = 9 | 11 (9–15) | 15 (5–15) | – | 1 | 1 | – | 7 |
| I° n = 7 | 15 (11–15) | 15 (15–15) | – | 1 | 1 | – | 5 |
| II° n = 1 | 7 | 15 | – | – | – | – | 1 |
| III° n = 1 | 15 | 15 | – | – | – | – | 1 |
| Significance of the difference: no IVH vs all IVH cases | 1.03; 0.304 0.3515 0.0000 (0–4) | 0.52; 0.602 0.9334 0.0000 (0–0) | | | | | |
| Hyperbilirubinemia, n = 14 | 11 (6–15) | 9 (4–15) | – | 1 | 3 | 3 | 7 |
| Significance of the difference: born at term with or without hyperbilirubinemia | **2.88; 0.004** 0.0521 0.000 (0–4) | **2.88; 0.004** 0.0059 3.000 (0–7) | | | | | |
| Hypotrophy, n = 10 | 15 (15–15) | 15 (15–15) | – | – | 1 | 1 | 8 |
| Significance of the difference: born at term with vs without hypotrophy | 0.11; 0.914 0.1744 0.000 (0–0) | 0.32; 0.752 0.3151 0.000 (0–0) | | | | | |
| Preterm+ hyperbilirubinemia, n = 10 | 10 (1–15) | 10 (9–15) | – | 1 | 2 | 1 | 6 |
| Significance of the difference: born preterm with vs without hyperbilirubinemia | **2.36; 0.018 0.0191** 2.000 (0–8) | 1.48; 0.138 0.1805 0.000 (−6 to 0) | | | | | |
| Significance of the difference: born at term with hyperbilirubinemia vs born preterm with hyperbilirubinemia | 0.39; 0.694 0.6809 0.000 (0–0) | 0.53; 0.598 0.6012 0.000 (0–0) | | | | | |
| Preterm+IVH+RDS, n = 10 | 5 (0–15) | 10 (0–15) | 3 | 1 | 2 | – | 4 |
| Preterm+RDS, n = 6 | 4;8;8;15;15;15 | 4;12;12;15;15;15 | – | 1 | – | 1 | 4 |
| Preterm+IVH, n = 6 | 6;7;9;11;15;15 | 6;6;9;11;15;15 | – | 1 | 1 | 1 | 3 |
| Preterm+IVH+ hyperbilirubinemia, n = 5 | 0;0;2;12;15 | 0;2;7;15;15 | – | – | 3 | – | 2 |
| Preterm+IVH+RDS+ hyperbilirubinemia, n = 4 | 0;0;0;15 | 0;0;0;11 | 1 | – | 1 | – | 2 |

(Continued)

| The risk factors | Quality in the prone position Median (Q25–Q75) | Quality in the supine position Median (Q25–Q75) | Final assessment at the age of 9 months | | | | |
|---|---|---|---|---|---|---|---|
| | | | Suspected CP | 6th month | 7th month | 8th month | 9th month |
| Preterm+hypotrophy, n = 3 | 2;0;0 | 0;6;6 | 1 | – | – | 1 | 1 |
| Preterm+IVH+hypotrophy n = 2 | 7;7 | 6;6 | – | 1 | – | 1 | – |
| RDS+hyperbilirubinemia, n = 2 | 7;15 | 4;15 | – | – | – | – | 2 |
| IVH+RDS, n = 1 | 0 | 0 | – | – | – | – | 1 |
| Preterm+hypotrophy+ hyperbilirubinemia, n = 1 | 0 | 0 | – | – | 1 | – | – |
| Preterm+IVH+hypotrophy+hyperbilirubinemia, n = 1 | 5 | 4 | – | – | 1 | – | – |
| Preterm+RDS+hypotrophy+hyperbilirubinemia, n = 1 | 0 | 0 | – | 1 | – | – | – |

**Table 2 The impact of risk factors on individual elements of motor performance was studied in the 3rd month in the prone position.** For each pair of variables, the values of Cramer's V coefficient, confidence interval, and Goodman and Kruskal Tau coefficient are given, along with the exact $p$-value. The ordered log it analysis was used to assess the particular elements' impact on the motor performance in the ninth month. The dependent variable was measured in the ordinal scale, while all other predictors were expressed in the nominal (binary) scale. For the significant models ($p < 0.05$), the pseudo R2 value along with $p > [z]$ was given. Only significant value are shown.

| Qualitative characteristics in the prone position: | Side of the body; right = R, left = L | Cramer's V, G-K-Tau, $p$ = | | | Ologit |
|---|---|---|---|---|---|
| | | Apgar 5th minute lower than 8 | IVH | hyperbilirubinemia | The impact on the motor performance in the ninth month, $p > [z]$ |
| Isolated head rotation | | 0.1782 (0.0718–0.2846); 0.0318; $p = 0.0070$ | 0.1984 (0.0917–0.3952); 0.0394; $p = 0.0001$ | 0.1676 (0.0610–0.2742); 0.0281; $p = 0.0010$ | 0.231 |
| Arm in front, forearm in an intermediate position, elbow outside of the line of the shoulder | R | 0.2101 (.01220–0.2981); 0.0189; $p < 0.0001$ | 0.1792 (0.0804–0.2779); 0.0321; $p = 0.0004$ | 0.1202 (0.0279–0.2285); 0.0164; $p = 0.0123$ | 0.153 |
| | L | 0.2170 (0.1278–0.3062); 0.0471; $p < 0.0001$ | 0.2248 (0.1274–0.3222); 0.0505; $p < 0.0001$ | 0.1371 (0.0360–0.2383); 0.0188; $p = 0.0069$ | 0.269 |
| Palm loosely open | R | 0.1988 (0.0726–0.3250); 0.0395; $p = 0.0003$ | 0.2590 (0.1364–0.3816); 0.0671; $p < 0.0001$ | 0.2090 (0.0888–0.3292); 0.0437; $p = 0.0001$ | 0.992 |
| | L | 0.1687 (0.0434–0.2948); 0.0285; $p = 0.0018$ | 0.2813 (0.1587–0.4040); 0.0791; $p < 0.0001$ | 0.2090 (0.0888–0.3292); 0.0437; $p = 0.0001$ | 0.992 |

| Table 2 (continued) | | | | | |
| --- | --- | --- | --- | --- | --- |
| Qualitative characteristics in the prone position: | Side of the body; right = R, left = L | Cramer's V, G-K-Tau, p = | | | Ologit |
| | | Apgar 5th minute lower than 8 | IVH | hyperbilirubinemia | The impact on the motor performance in the ninth month, p > [z] |
| Thumb outside | R | 0.0707 (0.0231–0.3002); p = 0.0261 | 0.2210 (0.0873–0.3546); 0.0488; p = 0.0010 | 0.2710 (0.1368–0.4052); 0.0734; p < 0.0001 | 0.990 |
| | L | 0.1243 (−0.0080 to 0.2567); 0.0155; p = 0.0224 | 0.2429 (0.1084–0.3773); 0.0833; p < 0.0001 | 0.2667 (0.1333–0.4002); 0.0711; p < 0.0001 | 0.991 |
| Spine segmentally in extension | | 0.1244 (0.0222–0.2267); 0.01550; p = 0.0187 | 0.1972 (0.0965–0.2979); 0.0389; p = 0.0001 | 0.1992 (0.0982–0.3001); 0.0397; p = 0.0001 | 0.407 |
| Scapula situated in the medial position: | R | 0.2114 (0.1232–0.2997); 0.0447; p<0.0001 | 0.1992 (0.1013–0.2971); 0.0379; p = 0.0001 | 0.2168 (0.1194–0.3142); 0.0470; p < 0.0001 | 0.354 |
| | L | 0.2199 (0.1301–0.3096); 0.0483; p < 0.0001 | 0.1916 (0.0916–0.2917); 0.0367; p = 0.0001 | 0.1408 (0.0393–0.2423); 0.0198; p = 0.0047 | 0.086 |
| Pelvis in the intermediate position | | 0.1235 (0.0106–0.2364); 0.0153; p = 0.0163 | 0.1732 (0.0617–0.2847); 0.0300; p = 0.0008 | 0.2610 (01500–0.3720); 0.0681; p < 0.0001 | 0.446 |
| Lower limbs situated loosely on the substrate | R | 0.1239 (−0.0012–0.2490); 0.0154; p = 0.0200 | 0.1919 (0.0681–0.3158); 0.0368; p = 0.0040 | 0.2340 (0.1094–0.3585); 0.0547; p < 0.0001 | 0.138 |
| | L | 0.1239 (−0.0012 to 0.2490); 0.0154; p = 0.0200 | 0.1919 (0.0681–0.3158); 0.0368; p = 0.0040 | 0.2561 (0.1309–0.3813); 0.0656; p < 0.0001 | 0.245 |
| Foot in intermediate position | R | 0.1515 (0.0117–0.3039); 0.0249; p = 0.0067 | 0.1752 (0.0408–0.3097); 0.0307; p = 0.0017 | 0.2613 (0.1219–0.4007); 0.683; p < 0.0001 | 0.126 |
| | L | 0.1578 (0.0081–0.2950); 0.0230; p = 0.0085 | 0.1833 (0.046–0.3199); 0336; p = 0.0012 | 0.2714 (0.1301–0.4126); 0.0736; p < 0.0001 | 0.599 |

**Table 3 The impact of risk factors on individual elements of motor performance was studied in the 3$^{rd}$ month in the supine position.** For each pair of variables, the values of Cramer's V coefficient and confidence interval and Goodman and Kruskal Tau coefficient are given, along with the exact $p$-value. The ordered log it analysis was used to assess the particular elements' impact on the motor performance in the ninth month. The dependent variable was measured in the ordinal scale, while all other predictors were expressed in the nominal (binary) scale. For the significant models ($p < 0.05$), the pseudo R2 value along with $p > [z]$. Only significant values are shown.

| Qualitative characteristics in supine position: | Side of the body, right = R, left = L | Cramer's V, G-K-Tau, p = | | | Ologit |
|---|---|---|---|---|---|
| | | Apgar 5$^{th}$ minute lower than 8 | IVH | hyperbilirubinemia | The impact on the motor performance in the ninth month, $p > [z]$ |
| Head symmetry | | 0.1380 (0.0274–0.02486); 0.0190; $p = 0.0077$ | 0.1958 (0.0865–0.3051); 0.0383; $p = 0.0002$ | 0.1473 (0.0392–0.2554); 0.0217; $p = 0.0039$ | 0.134 |
| Spine in extension | | 0.1244 (0.0222–0.2267); 0.0155; $p = 0.0187$ | 0.1972 90.0965–0.2979); 0.0389; $p = 0.0001$ | 0.1992 (0.0982–0.3001); 0.0397; $p = 0.0001$ | 0.633 |
| Shoulder in balance between external and internal rotation | R | 0.2196 (0.1122–0.3269); 0.0482; $p = 0.0000$ | 0.1979 (0.0883–0.3075); 0.0392; $p = 0.0001$ | 0.2059 (0.0967–0.3152); 0.0424; $p = 0.0001$ | 0.014 |
| | L | 0.0214 (0.1081–0.3205); 0.0459; $p = 0.0001$ | 0.2111 (0.1026–0.3196); 0.0446; $p = 0.0001$ | 0.2364 (0.1286–0.3443); 0.0559; $p = 0.0000$ | 0.168 |
| Wrist in intermediate position | R | 0.2194 (0.0693–0.3696); 0.0482; $p = 0.0003$ | 0.2499 (0.1099–0.3998); 0.0624; $p < 0.0010$ | 0.2517 (0.1142–0.3893); 0.0634; $p < 0.0001$ | 0.310 |
| | L | 0.1762 (0.0332–0.3202); 0.0310; $p = 0.0026$ | 0.2590 (0.1544–0.4358); 0.0870; $p < 0.001$ | 0.2427 (0.1069–0.3784); 0.0589; $p < 0.0001$ | 0.238 |
| Thumb outside | R | 0.1516 (0.0081–0.2958); 0.0230; $p = 0.0085$ | 0.1473 (0.0165–0.2780); 0.0217; $p = 0.0067$ | 0.2347 (0.0970–0.3724); 0.0551; $p < 0.0001$ | 0.792 |
| | L | 0.147 (0.0029–0.2825); 0.0204; $p = 0.0117$ | 0.2182 (0.0812–0.3552); 0.0476; $p = 0.0001$ | 0.2730 (0.1350–0.4110); 0.0745; $p < 0.0001$ | 0.034 |
| Palm in intermediate position | R | 0.1086 (−0.0251 to 0.2422); 0.0118; $p = 0.0423$ | 0.2224 (0.0845–0.3604); 0.0495; $p = 0.0001$ | 0.2256 (0.0898–0.3614); 0509; $p = 0.0001$ | 0.955 |
| | L | 0.1036 (−0.0279 to 0.2350); 0.0107; $p = 0.0507$ | 0.2681 (0.1285–0.4076); 0.0719; $p < 0.0010$ | 0.2427 (0.1069–0.3784); 0.0589; $p < 0.0001$ | 0.221 |
| Pelvis extended (no anteversion, no retroversion) | | 0.1542 (0.0396–0.2686); 0.0237; $p = 0.0036$ | 0.1777 (0.0654–0.2899); 0.0316; $p = 0.0006$ | 0.2856 (0.1745–0.3968); 0.0816; $p = 0.0000$ | 0.165 |
| Lower limb situated in moderate external rotation | R | 0.2233 (0.0901–0.3546); 0.0499; $p = 0.0001$ | 0.1980 (0.0728–0.3231); 0.0392; $p = 0.0001$ | 0.1959 (0.0724–0.3194); 0.0384; $p = 0.0003$ | 0.083 |
| | L | 0.2181 (0.0864–0.3497); 0.0475; $p = 0.0000$ | 0.1919 (0.0681–0.3158); 0.0368; $p = 0.0004$ | 0.2118 (0.0883–0.3354); 0.0449; $p = 0.0001$ | 0.115 |

| Table 3 (continued) | | | | | |
|---|---|---|---|---|---|
| Qualitative characteristics in supine position: | Side of the body, right = R, left = L | Cramer's V, G-K-Tau, $p$ = | | | Ologit |
| | | Apgar 5[th] minute lower than 8 | IVH | hyperbilirubinemia | The impact on the motor performance in the ninth month, $p > [z]$ |
| Lower limb bent at a right angle at hip and knee joints, foot in intermediate position – lifting above the substrate | R | 0.2009 (0.0740–0.3278); 0.0404; $p$ = 0.0004 | 0.2196 (0.0947–0.3391); 0.0470; $p$ = 0.0001 | 0.3185 (0.1965–0.4404); 0.1014; $p$ = 0.0000 | 0.328 |
| | L | 0.1966 (0.0709–0.3222); 0.0386; $p$ = 0.0005 | 0.2114 (0.0903–0.3326); 0.0447; $p$ = 0.0001 | 0.3118 (0.1908–0.4327); 0.0972; $p$ = 0.0000 | 0.311 |

study were patients/clients of the Child Neurology Center. All parents/caregivers written agreed to participate in the study, as apart from routine assessment and therapy, no extra visit was necessary.

### Statistics

Due to the nature of the variables, the results were presented as median with quartiles (Me, Q25–Q75), with an assumed level of significance $p < 0.05$, two-tailed test. To compare two groups, the Mann-Whitney U test was used, to compare more than two groups: Kruskal-Wallis ANOVA and *post hoc* Dunn's test. To estimate the effect size, the Hodges-Lehman estimation of the Wilcoxon-Mann-Whitney test from StatXact-11 Cytel Studio v.11.1.0 was used.

The assumed statistical significance level was $p < 0.05$.

The association between pairs of nominal categorical variables was tested using the following tests:

1) To measure the magnitude of the association between two nominal variables without regard to the dimensions of the r x c contingency table, Cramer's V coefficient was used in place of Pearson's chi-square statistics; the higher the coefficient, the stronger the association.

2) To measure the proportion of variation between interrelated nominal variables, the Goodman-Kruskal's Tau test was used; the higher the result, the more substantial the influence of one variable on another.

3) To estimate the best predictors of the impact of particular risk factors on the final assessment in the 9[th] month of life, the ordered logit analysis was performed.
The dependent variable was measured on the ordinal scale, while all other predictors were expressed on the binary scale.

For the whole model, the pseudo $R2$ value and likelihood ratio was given and $P > [z]$ value was shown for each element.

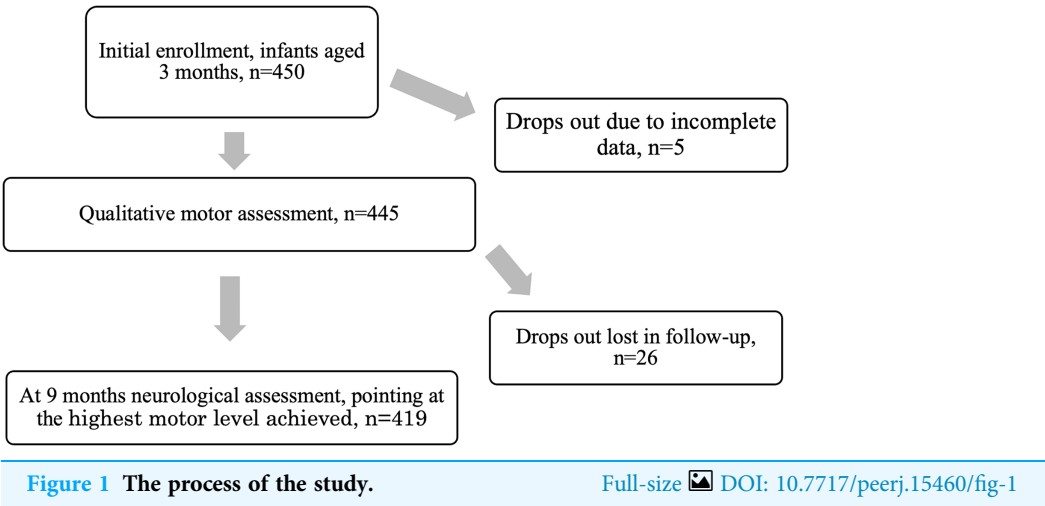

**Figure 1 The process of the study.**

For both the association tests and ordered logit test exact probability values of the two-sided $p$ (instead of asymptotic $p$-values) were calculated using StatXact-11 Cytel Studio v.11.1.0.

The assumed statistical significance level was $p < 0.05$.

## RESULTS

In our previous studies sex differences were never present (*Gajewska et al., 2013*; *Gajewska, Sobieska & Moczko, 2014*). Also in this study, the difference between boys and girls was not statistically significant (see Table 1). When children were divided according to prematurity and then according to sex, following results were obtained: boys premature, $n = 71$, prone 13 (5–15); supine 15 (6–15); girls premature, $n = 58$, prone 15 (8–5); supine 15 (6–15); boys born at term, $n = 165$, prone 15 (10–15), supine 15 (13–15); girls born at term, $n = 125$, prone 15 (9–15), supine 15 (12–15). Any difference was statistically significant.

### Qualitative assessment at the age of 3 months

The assessment at the age of 3 months is expressed as the sum of elements, in the prone and supine positions (enumerated in Tables 2 and 3, respectively), with a maximum value of 15 points. Exact data for the sum in the prone and supine positions are given in Table 1. For each subset of the investigated group, divided according to various criteria, the median with quartiles for the sum in prone position and sum in supine position is given, as assessed by the physiotherapist in the $3^{rd}$ month of life. Next, each subset of the investigated group the number of children is divided according to the maximal level of development they achieved in the $9^{th}$ month of life, as assessed by the neurologist. The particular subsets are compared and the statistical results are depicted for each comparison. The sum in both positions decreased with the increasing number of risk factors and was always lower in preterm infants burdened with risk factors. Additionally, the sum in the prone position was usually lower than in the supine position.

## Impact of particular risk factors on motor performance at the age of 9 months

First, the impact of individual risk factors or their combination on the qualitative assessment of motor skills in the 3[rd] month of life and the maximum skill level achieved at 9 months were investigated. The details of this analysis are presented in Table 1. However, it should be noted that there were no children with three or four risk factors that were not born prematurely.

When children were divided according to the fifth-minute-Apgar category, the sum in the prone position and the sum in the supine position were significantly different. For the prone position, KW ANOVA showed high significance of the difference ($H = 14.19$, $p = 0.001$; *post hoc* test: Apgar 4–7 *vs* Apgar 8–10, $z = 2.88$; $p = 0.012$; Apgar 0–3 *vs* Apgar 4–7, $z = 0.58$; $p = 1.000$; Apgar 0–3 *vs* Apgar 8–10, $z = 1.73$, $p = 0.252$). For the supine position, there was also a significant difference between the three groups ($H = 12.46$; $p = 0.002$). However, the *post hoc* test did not point at any significant difference (Apgar 4–7 *vs* Apgar 8–10, $z = 1.09$; $p = 0.830$; Apgar 0–3 *vs* Apgar 4–7, $z = 2.02$; $p = 0.129$; Apgar 0–3 *vs* Apgar 8–10, $z = 2.25$, $p = 0.074$).

As the strength of the test was very low when all three Apgar categories were compared, with the smallest group (Apgar 0–3, $n = 2$), the exact comparison was made for between the categories Apgar 4–7 *vs* Apgar 8–10 and it showed a statistically significant difference (see Table 1).

There were altogether 38 children with hyperbilirubinemia but only in ten born at term and in ten preterm infants it was a single risk factor. The difference between children born at term without any risk factors *vs* those born at term with hyperbilirubinemia was statistically significant, the difference between children born preterm with *vs* without hyperbilirubinemia was significant only in the prone position. The difference between children born with hyperbilirubinemia at term *vs* preterm was not significant.

There were altogether 18 children born with hypotrophy but only in ten born at term it was a single risk factor. The difference between children born at term without any risk factors *vs* those born at term with hypotrophy was not significant (see Table 1).

All children with a suspicion of CP in the 9[th] month of life did not perform correctly in the 3[rd] month. This diagnosis was confirmed at 18 months (nine participants were diagnosed with tetraplegia and one with diplegia). Children with tetraplegia failed to perform any of the evaluated elements in the prone and supine positions (scored 0 points), and only one child finally diagnosed with diplegia (weight 3,210 g; born on the 40[th] week of gestation) scored 6/15 points in the supine position (hands, thumbs, external rotation of the lower limbs). Out of 10 children with suspicion of CP, four were affected by grade I IVH, one with grade II IVH, five with RDS, one with hyperbilirubinemia, and one with hypotrophy. When the total number of risk factors was taken into account, four children finally diagnosed with CP were not affected by any risk factor, one suffered from hypotrophy, four were affected by two risk factors (IVH + RDS), and one was affected by three (IVH + RDS + hyperbilirubinemia) (see Table 1).

The type of delivery (vaginally, $n = 224$; Caesarean section, $n = 158$; forceps delivery, $n = 23$; vacuum, $n = 14$) did not entail significant differences in the 3rd month of life or the maximum development at 9 months of age (KW ANOVA, H = 2.54; $p = 0.468$ for prone position and H = 7.19; $p = 0.066$ for the supine position).

Children born prematurely and at term, without risk factors, achieved a similar maximum development level. Almost half of the children born prematurely and with additional risk factors showed delayed motor development and CP in extreme cases. In contrast, most children born at term but affected by risk factors still achieved the proper level of motor development (see Table 1).

It seems that prematurity does not cause a significant delay in motor development. Still, in combination with risk factors, IVH, RDS, and hyperbilirubinemia, it results in a notably worse motor development prognosis (see Table 1).

Proper development manifests in achieving all the postural and motor characteristics listed in the scale, *i.e.*, the maximum sum of points. Its reduction indicates deficits/abnormalities and may herald/predict developmental delay. Therefore we have evaluated whether risk factors in infants could coincide with reduced assessment scores. The effect of risk factors on the total qualitative assessment in the prone and supine positions was analyzed.

The result of the detailed qualitative assessment of motor performance was first presented in the form of a sum of points obtained to demonstrate to what extent individual risk factors or their combination reduces the level of functioning. The results of this analysis are presented in Table 1. The final development level of children with a given risk factor or their combinations in the 9th month of life was also indicated. Then, we investigated which risk factors influenced specific motor components assessed in pronation and supination. The results of this analysis are presented in Tables 2 and 3, respectively.

The last column of these tables indicates which motor element assessed in the 3rd month of life was critical (had a significant effect) to the child's achievement of the maximum level of motor development assessed at 9 months of age. Statistical significance suggests that given risk factors disturbed the correct position or function in the examined children at the third month of life.

The differences of the "sum in the prone position" (H = 49.08, $p = 0.000$) and "sum in supine position" (H = 47.69, $p = 0.000$) variables depending on the number of risk factors were statistically significant, while the difference investigated using the *post hoc* test was as followed: without risk factors/two risk factors $p = 0.002$; without risk factors/three risk factors $p = 0.003$. Depending on the type of risk factors in the studied children, it was also possible to demonstrate the differences in the "sum in the prone position" (H = 52.81, $p = 0.004$) and "sum in supine position" (H = 13.21, $p = 0.022$) variables were significant, but detailed comparisons failed to achieve significance.

Next, the influence of particular risk factors on motor elements, investigated in the prone and supine positions, was studied. Six risk factors were included in the analysis: prematurity, 5th min-Apgar score 4–7, the presence of IVH, RDS, hyperbilirubinemia, and

hypotrophy. Only reduced 5[th] min Apgar score, IVH and hyperbilirubinemia were repeatedly significant and were presented in Tables 2 and 3.

Subsequently, we analyzed which elements of motor skills, assessed qualitatively in the prone and supine positions in the 3[rd] month of life, had the most significant impact on the maximal level of motor performance, evaluated by the neurologist at the age of 9 months. The axial features, namely the spine, scapulae, and pelvis, showed the highest impact. The results of this analysis are listed in detail in Tables 2 and 3.

The elements in the prone position which best determined the proper prognosis of motor development included: correct curvatures of the vertebral column, scapula situated in the medial position, pelvis in the intermediate position, lower limbs situated loosely on the rehabilitation table, while in the supine position these comprised: proper curvatures of the vertebral column, shoulder in a balance between external and internal rotation, pelvis extended, lower limbs bent at a right angle at hip and knee joints, foot in the intermediate position—lifted above the rehabilitation table. The ologit analysis showed the pseudo $R^2$ value of 0.4023 and the likelihood ratio of 267.02. Only for few elements the model showed significant $P > (z)$. The results are presented in the Tables 2 and 3, last column.

## DISCUSSION

Some authors point out that the early detection of motor abnormalities is relatively difficult (*Crnković et al., 2011*). Thus, it seems advisable to examine diagnostic methods that would make it possible to detect children at risk of abnormal development at the earliest possible time, irrespective of the risk factors involved. We present an assessment method that was validated in comparison to neurologic check, it is short, relatively easy to follow and allows to depict particular elements of motor performance at the crucial age of 3 months (*Gajewska et al., 2013*; *Gajewska, Sobieska & Moczko, 2014*; *Gajewska et al., 2021a*).

The primary finding is that none of the singular risk factors universally recognized as the most dangerous (prematurity, IVH, RDS, hyperbilirubinemia) is responsible for severe motor development impairment. However, the higher the number of coinciding risk factors, the worse the prognosis for motor development.

The percentage of premature births is 7.1% in Europe (Caring for tomorrow– EFCNI) and approximately 6.5% in Poland (*Główny Urząd Statystyczny, 2011*). It was shown that RDS, IVH and sepsis, hypoglycemia, hypernatremia, and hypothermia are the factors causing developmental delay and potentially leading to unfavorable long-term neurodevelopmental consequences (*Khan et al., 2012*; *Stephens & Vohr, 2009*).

One risk factor did not worsen the quality in the children born at term. Preterm delivery as a single risk factor significantly affected motor performance, though the median quality did not differ. Yet, the relative risk of CP was higher in those children (3/68 = 4.4% *vs* 1/211 = 0.5% in born at term). The more risk factors, the lower the quality. The difference was significant for three *vs* fewer risk factors in the supine position. However, it should be noted that there were no children with an extremely low gestational age in the investigated group who could exhibit a delay despite being examined at the corrected age.
Many authors report that the CP diagnosis is often made too late, and rehabilitation, which could help improve the affected children's condition, is implemented with a significant delay (*Morgan et al., 2015*). *Novak et al. (2017)* identified two diagnostic pathways for infants at risk of developing CP. About 50% are infants from the risk group with certain factors, such as prematurity, fetal growth disorders, encephalopathy, genetic defects, and convulsions, and are diagnosed under 5 months of age (corrected age). Infants with no such medical/ clinical history are diagnosed later in life, based on the second pathway. The first disturbing symptoms in such infants include a delay in motor development (*e.g.*, lack of sitting ability at 9 months of age, or a clear one-sided preference, visible only when performing more complex actions (*e.g.*, griping) (*Novak et al., 2017*)).

It is worth noting that *Novak et al. (2017)* suggest diagnostics only after the age of 5 months, while in our studies (repeatedly), children who were diagnosed with CP at the age of 18 months showed large motor deficits at 3 months of age (they did not perform any, or only performed 2–3 activities of the assessed 15 in pronation and supination) (*Gajewska et al., 2013*; *Gajewska, Sobieska & Moczko, 2014*; *Gajewska et al., 2015a*).

However, early detection of motor deficits may support therapy for children at risk of CP and those who will eventually only develop a developmental delay. Moreover, in the third month of life, it could be noticed that they did not perform all of the observed functions correctly, or at least did not achieve the maximum score. Hence, we suggest that diagnostics should be performed very early, already at 3 months. Even if the diagnosis of CP may be delayed until 18 months, rehabilitation should be implemented as early as any worrying symptoms are noticed.

The most crucial aim of the study was to demonstrate whether it is possible to associate specific risk factors with a motor delay up to the occurrence of CP.

As for particular risk factors, an Apgar score lower than eight significantly worsened both the quality in children aged 3 months and the prognosis of further motor development and the relative risk of CP (3/12 = 25% *vs* 7/394 = 1.8%). IHV I° was not a thread, and the number of children with higher grades was too low to perform the analysis. Hyperbilirubinemia significantly affected motor performance in 3-month-old children, and in half of them caused a delay in motor development, and it affected equally children born at term and preterm. It should be regarded as a potential threat to further motor development even in children without any other risk factors and these children should be followed, carefully supervised by a physiotherapist, and subjected to proper therapy if necessary. In contrast, hypotrophy did not cause any delay in motor development if present as a single risk factor. Other investigated risk factors were present in too few children for exact analysis. However, an individual motor performance at 3 months was low, and motor development was delayed.

It is also worth emphasizing that these factors, acting not individually but in combination, had the most significant effect on motor development delay.

Apart from the statement that children with particular risk factors developed more slowly, it was shown that their performance in the 3rd month was worse. Only a detailed qualitative assessment can detect these minor deviations from normal development that affect motor progress. At the same time, detection of disorders in the 3rd month of life

allows for the implementation of early physiotherapy, following commonly accepted canons.

The sum in the prone position score seems to reflect better the discrete deficits seen in children born with a poorer 5[th] min Apgar score. The ability to overcome the forces of gravity is probably a good indicator of both the proper development of muscle strength and the maturity of the nervous system.

The predictive value of the commonly used Bayley scale is being undermined (highly unstable delay classifications, low sensitivities, and poor positive predictive values), and the need for a new, more effective tool used to predict motor development and allow early therapeutic intervention is emphasized (*Lobo et al., 2014*).

Since there is no globally recognized "gold standard" method of functional assessment, it was impossible to compare the results of this study with such a standard or to calculate the confidence intervals or odds ratio. The only point of reference was the neurological examination, and more specifically, the level of motor development assessed at the age of 9 months. This age was chosen as this is the usual time for assuming the standing position (*Vojta & Peters, 2007*; *Gajewska, Sobieska & Moczko, 2014*). Achieving this milestone (with the support of furniture or an adult) reflects the achievement of complete motor control and guarantees further proper motor development (including walking) (*Adolph & Hoch, 2019*).

Another important risk factor for developmental disorders and CP is the increasing severity of IVH (*Fily et al., 2006*; *Spittle et al., 2009*). Measures of brain structure and function are by far the most predictive of neurodevelopmental outcomes. Preterm infants with ventricular dilatation and IVH showed worse motor test results than those without IVH (*Vollmer et al., 2006*). IVH in children born prematurely leads to worse psycho-motor assessment outcomes and more frequent CP occurrence (*Klebermass-Schrehof et al., 2012*). In our study, in the group of children who obtained bad scores, IVH was much more frequent. In the studies by *Sherlock, Anderson & Doyle (2005)*, patients with IVH grade IV showed up to four times higher percentages of abnormal results than grade I patients. However, we could not confirm this finding due to the small number of children with higher grades of IVH included in our study.

The risk of the RDS occurrence is inversely proportional to the newborn's gestational age- occurs in 1% of all newborns and nearly 70% of infants born before the 28th week of gestation (*Shonkoff & Meisels, 2000*). In our study, RDS occurred mainly in children whose motor development was assessed as inferior. However, it should be stressed that these children were not born extremely premature.

The analysis of risk factors and their impact on motor development in the investigated group was similar to that of other authors. Some papers put more attention to socio-economic risk factors (*Sania et al., 2019*) other relate to the combination of social and biologic risk factors, but investigate gross motor function and cognitive functions, whereas we decided to focus on biological factors only and investigate fine motor skills in details (*Tian et al., 2018*). It is believed that there is a critical need for collaboration among experts to determine early predictive factors and neuroprotective therapies (*Khan et al., 2012*). Furthermore, while hyperbilirubinemia proved to be a highly burdening factor, even

children who suffered from many complications of similar severity occasionally showed proper development and reached maximal performance at 9 months.

The qualitative assessment conducted at the age of 3 months is a reliable prognosis of motor development at 9 months of age, with a crucial role played by proximal characteristics related to the axial skeleton (spine-scapulae-shoulders-pelvis). Ologit analysis implies the independent predictors that affect the prognosed variable. It should be noticed that the particular elements of the motor performance in the third month are in fact related to each other. That is probably why the whole model testing the impact of those elements on the level of the motor development achieved in the ninth month showed relatively high likelihood, though particular elements did not show the statistically significant relation. In other words, all elements in the third month are necessary for the perfect motor performance in the ninth month, and the more central elements are disturbed, the more delay one can expect.

One can notice that the values of Cramer's V coefficient were not high (the highest value observed was 0.2856); the same can be seen for the Goodman and Kruskal Tau coefficient (the highest value was 0.0972). The highest values were observed for hyperbilirubinemia.

Thus, the presence of risk factors affects motor performance as a whole, but some elements seem more critical. The axial elements (spine-scapulae-pelvis-lower limbs) showed more pronounced dependence—particularly for hyperbilirubinemia. The precise determination of which motor development elements are impaired also allows for the implementation of appropriate therapy.

Four children whose diagnosis of tetraplegia was ultimately confirmed at the age of 18 months were not affected by any risk factors but scored zero during the quantitative assessment in the 3rd month, both in prone and supine positions. Qualitative and quantitative evaluation makes it possible to focus on motor delays or disorders as early as in the 3rd month of life. According to *McIntyre et al. (2011)*, 50% of CP cases are diagnosed in infants born at term in whom no risk factor has been identified.

## STRENGTHS AND LIMITATIONS

The greatest limitation is that no extrinsic constraints were evaluated. We also confirm statistical limitations due to the number of cases or absence of cases in some cells. Also, the problem of the absence of a gold standard instrument is an external limitation.

The presented paper is based on a large and homogenous study group, and the implemented statistical analysis, not commonly used in similar research, is accordingly adjusted to the hypothesis. Qualitative analysis was performed in the third month of life, regarded as a crucial time point to predict further development, allowing to plan therapy or social support in cases of expected disability.

A relatively short follow-up (up to 9 months) is the only limitation of the study.

## CONCLUSIONS

It was not prematurity itself but its combination with risk factors (IVH, RDS, hyperbilirubinemia) that made the prognosis of proper motor development worse. Children with a motor delay at 9 months of age demonstrated a lower quality of movement

as early as in the 3$^{rd}$ month of life. Furthermore, qualitative assessment allowed to identify high-risk children and predict the degree of delay. Axial skeleton characteristics (vertebral column, scapulae, shoulders, pelvis) were the best determinants of the proper prognosis of motor development.

### Funding
The authors received no funding for this work.

### Competing Interests
The authors declare that they have no competing interests.

### Author Contributions
- Ewa Gajewska conceived and designed the experiments, performed the experiments, analyzed the data, prepared figures and/or tables, authored or reviewed drafts of the article, and approved the final draft.
- Jerzy Moczko conceived and designed the experiments, analyzed the data, prepared figures and/or tables, and approved the final draft.
- Mariusz Naczk analyzed the data, authored or reviewed drafts of the article, and approved the final draft.
- Alicja Naczk analyzed the data, authored or reviewed drafts of the article, and approved the final draft.
- Magdalena Sobieska conceived and designed the experiments, analyzed the data, prepared figures and/or tables, authored or reviewed drafts of the article, and approved the final draft.

### Human Ethics
The following information was supplied relating to ethical approvals (*i.e.*, approving body and any reference numbers):

The study was conducted at the Center for Child and Adolescent Neurology Clinic between 2018 and 2021, following the ethical guidelines of the 1964 Helsinki declaration and its later amendments. Children recruited for the study were patients/clients of the Child Neurology Center. All parents/caregivers agreed to participate in the study, as apart from routine assessment and therapy, no extra visit was necessary. The study was approved by the Research Ethics Committee of Poznan University of Medical Sciences and registered under no. 22/10 (07-01-2010).

### Data Availability
The dataset is available in the Supplemental Files.

## Supplemental Information

Supplemental information for this article can be found online at http://dx.doi.org/10.7717/peerj.15460#supplemental-information.

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
