# Peer review of "Impact of selected risk factors on motor performance in the third month of life and motor development in the ninth month"

_PeerJ, doi:10.7717/peerj.15460_

## Round 0.1 · original submission · Major Revisions

Although the articles present some merits, there are major concerns regarding the experimental design and validity of the findings that must be improved.

·

Basic reporting

Article needs English review and correction of ambiguous information. References are pertinent and well selected, but some statements need literature support. Some limitations in the recognition of partial approach of the problem, however in line with main purpose of the study. Tables need to be reviewed in content. Figure and tables were constructed with the purpose of presenting relevant information. All appropriatte raw data are available. Text has internal articulation and is self-contained.Suggested improvements in files attached.

Experimental design

Data seem honest and study main purpose can be considered covered by aims and scope of the journal. Research question is weel defined, is relevant and meaningful. Knowledge gap is identified and study contributes are identied and are pertinent. Instruments are well selected according main objective and sample age. Ethical standards are fullfilled. Methods are described with sufficient information to be reproducible. Instrument limitations are assumed.Suggested improvements in files attached.

Validity of the findings

The study reinforces previous studies and adds value to the literature. Instruments used were validated for the population of the sample. Objectives are tested with adequate statistical treatment, however with insufficient tools (e.g., no effect sizes were estimated, and some data is better understandable also with percentages). Some data are not presented and should. Limitations must be better discussed. Conclusions make sense but is mandatory that inferential statistical treatment needs to be more robust and more controlled. Suggested improvements in files attached.

Additional comments

Suggested improvements in files attached.

Reviewer 2 ·

Basic reporting

After a careful reading of your article, I point out the following aspects that should improve:

- the article should include a more robust introduction, focusing on the problem that the authors have formulated. In the literature review, the authors only identify individual risk factors and never point to studies investigating the impact of cumulative risk factors on the child's motor development.
- The authors write the following: “Earlier publications mainly intended to demonstrate a proper analysis of the qualitative assessment of motor development and determine predictors of further motor development (lines 104-105); please, give examples of such publications/studies.
- The following formatting aspects should be corrected: citation of authors in the text, incomplete references, complete description of data in tables 2 and 3, and table headings with a more accurate description. Please see all the annotations I made in the body of the manuscript. There are many details to correct.
- Some spelling/grammatical errors were also pointed out in the body of the manuscript. English must be proofread by a fluent speaker or professional.

Experimental design

- As mentioned above, the review must be more profound and justify the problem of the formulated study.
- The content described in the methods section should be more detailed, namely the description of the assessment instruments used (see notes made in the body of the manuscript);
- Statistical procedures should also be more precise. For example, “Cramer.s V coefficient was used in place of Pearson.s chi-square statistics; the higher the coefficient, the stronger the association”(lines 189-190). This is an ambiguous interpretation. Please specify the cut-off values to consider a weak, medium or strong association according to a theoretical framework.

Validity of the findings

- The section that most disappointed me was the section on the presentation of results, which conditioned my review of the discussion and conclusion of the study. I think that this section must be revised as it presents a confusing sequence and sometimes without consistency with the results reported in tables 1, 2 and 3. Please see the comments I made in the body of the text.
- There should be greater rigor in the way they report results and statistical symbols;
- why do the authors not present the results that do not give significant results? does not contribute to knowledge? I think yes! in tables 2 and 3, empty cells must have numbers or some note; otherwise, it seems that the table is unfilled.

Additional comments

Although the study presents a current and pertinent topic, the article must undergo a major review to be published.

In general, I can say that the introduction should be more robust and adequate to justify the research question raised in the manuscript; The methodology does not provide sufficient details on the motor assessment scales. The results section should present a more objective narrative compatible with the results in tables 1, 2 and 3.
Practical implications and methodological suggestions for future studies should be pointed out.

Throughout the manuscript, I made many notes that should deserve the greatest attention on the part of the authors.

Annotated reviews are not available for download in order to protect the identity of reviewers who chose to remain anonymous.

---

## Round 0.2 · Major Revisions

Please consider the significant concerns of reviewers regarding statistics and basic reporting.

·

Basic reporting

Tables still with cells without information is confusing for readers, this presentation format is not adequate.

Experimental design

no further comments

Validity of the findings

effect sizes are mandatory, identify, estimate and present

Additional comments

if a test validation was published, why don't you mention it in discussion? including making some statements relative to protocol used in this study

Reviewer 2 ·

Basic reporting

- the introduction needs to be more robust and focused on the study problem. In response to my review, the authors respond that no studies assess the impact of risk factors on motor development in children. Through a search on PubMed, I found several Studies, and some deserved to be cited. https://pubmed.ncbi.nlm.nih.gov/?term=risk+factors+motor+development

On this subject, the authors in the discussion even mention the following:
“The analysis of risk factors and their impact on motor development in the investigated group was
similar to that of other authors (lines 343-444); Are there studies or not? What are these studies that corroborate the results found?

I continue with the opinion that the authors should give greater theoretical support in the introduction, namely, what is the theoretical approach that supports the study when they refer to the following: “Out of those, the biological ones are regarded as the most crucial factors for motor development” ( lines 122-123)
To answer this question, I advise you to read the following article: Hadders-Algra M. Early human motor development: From variation to the ability to vary and adapt. Neurosci Biobehav Rev. 2018 Jul;90:411-427. doi: 10.1016/j.neubiorev.2018.05.009. Epub 2018 May 9. PMID: 29752957.

- how multiple authors are cited in a citation must be revised (see notes made in the article)
- how statistical results are reported should be reviewed; Statistical symbols must be in italics (see notes made in the article)

Experimental design

As mentioned above, the review must be more profound and justify the problem of the formulated study.

Validity of the findings

- why do the authors not present the results that do not give significant results? I suggest again that the authors report all the results found in the table.

Additional comments

The article has undergone improvements. However, the authors should still correct some aspects suggested in the 1st and 2nd revisions. in my opinion, the literature needs to be more robust and focused on the study problem
There should be greater rigor in the way they report results.
Please see the notes made in the manuscript.

Annotated reviews are not available for download in order to protect the identity of reviewers who chose to remain anonymous.

---

## Round 0.3 · Major Revisions

The authors should add the effect size calculation, a paragraph about the scientific contribution before the objective of the study, and an additional one in the discussion about the importance of the study.

·

Basic reporting

Tables still confusing for readers, connection between legends information and tables content not easy.

Experimental design

no further comments

Validity of the findings

effect sizes are mandatory, identify, estimate and present

Additional comments

no further comments

Reviewer 2 ·

Basic reporting

The authors should only review the two citations underlined in yellow. It is not necessary to repeat the author and date.

Experimental design

No comment

Validity of the findings

no comment

Additional comments

The article has been greatly improved. The authors responded to the suggestions and corrections indicated.
I underline in yellow only two citation details that could be improved, so as not to repeat the author or the year.

Annotated reviews are not available for download in order to protect the identity of reviewers who chose to remain anonymous.

---

## Round 0.4 · Minor Revisions

Some aspects can be improved at the moment. Please address the suggestions of the reviewers.

·

Basic reporting

no further comments

Experimental design

except personal misunderstanding, effect sizes a still not presented

Validity of the findings

no further comments

·

Basic reporting

Introduction
Please format the reference Vojta V & Peters A, 2007 along the manuscript.

In a general comment for the introduction session, I would like to suggest reorganizing it, i.e. including each topic in each paragraph. In this sense, I suggest one paragraph for the risks identified, another with a brief explanation regarding the battery tests, another with other studies and the last one with the aim of the study. Something similar to facilitate the understanding the study background. In fact, considering the title, I would expect to read more about the risk factors in this session.


Material and methods
How many neurologists participated in the analysis?

Results
Line 227 – for me, this sentence makes no sense to be here. I suggest you change it to methods to justify your decision making.

Table 2 and 3 – what is the meaning of “ologit”?

Discussion
This session is clear and easy to follow.

Experimental design

The research question and implementation seems clear and replicable.

Validity of the findings

This topic has a great importance to the medical field.

Additional comments

This paper seems well conducted, although I suggested improvements in the introduction section. Besides that, the paper seems to have a great importance in the field.

---

## Round 0.5 · Minor Revisions

The reviewers required some improvements in the introduction.

·

Basic reporting

Same changes were made regarding my previous considerations.

Experimental design

No comment.

Validity of the findings

No comment

Additional comments

I believe that the introduction could benefit from improvements in writing and not just the elimination of paragraphs. Anyway, I think it's in better shape than the first version.
Another issue that I agree with my colleague reviewer is the inclusion of effect sizes, in fact, it was added on the table, but no discussion of the values was added.

---

## Round 0.6 · Major Revisions

The authors could not follow the suggestions of the different reviewers across the stage. Many issues are still lacking in methods:
Example: eligibility criteria and sampling strategy are not presented; the statistical assumptions for running the different tests are not presented, thus not confirming if the statistical data follow the requirements.

·

Basic reporting

I feel that despite the Authors indicating that there are valuable comments, the authors have no real intention to follow the comments. Already in the previous round I had felt that, since only a few changes had been made, such as deleting paragraphs. In this round I come across deleted text to be rewritten again (perhaps believing that the Reviewer would only look at the colors of the track changes). So this attitude leads me to suggest rejection.

Experimental design

Nothing to declare.

Validity of the findings

Nothing to declare.

Additional comments

Nothing to declare.

---

## Round 0.7 · accepted · Accept

The manuscript can be accepted.